# Antagonistic interaction between Ezh2 and Arid1a coordinates root patterning and development via Cdkn2a in mouse molars

Junjun Jing[1,2], Jifan Feng[1], Jingyuan Li[1], Xia Han[1], Jinzhi He[1,2], Thach-Vu Ho[1], Jiahui Du[1], Xuedong Zhou[2], Mark Urata[1], Yang Chai[1]*

[1]Center for Craniofacial Molecular Biology, Herman Ostrow School of Dentistry, University of Southern California, Los Angeles, United States; [2]State Key Laboratory of Oral Diseases, National Clinical Research Center for Oral Diseases, West China Hospital of Stomatology, Sichuan University, Chengdu, China

**Abstract** Patterning is a critical step during organogenesis and is closely associated with the physiological function of organs. Tooth root shapes are finely tuned to provide precise occlusal support to facilitate the function of each tooth type. However, the mechanism regulating tooth root patterning and development is largely unknown. In this study, we provide the first in vivo evidence demonstrating that Ezh2 in the dental mesenchyme determines patterning and furcation formation during dental root development in mouse molars. Mechanistically, an antagonistic interaction between epigenetic regulators Ezh2 and Arid1a controls Cdkn2a expression in the dental mesenchyme to regulate dental root patterning and development. These findings indicate the importance of balanced epigenetic regulation in determining the tooth root pattern and the integration of roots with the jaw bones to achieve physiological function. Collectively, our study provides important clues about the regulation of organogenesis and has general implications for tooth regeneration in the future.
DOI: https://doi.org/10.7554/eLife.46426.001

*For correspondence:
ychai@usc.edu

Competing interests: The authors declare that no competing interests exist.

## Introduction

Control of organ patterning is crucial for organ function and is a fundamental aspect of biology. Teeth are important for a number of physiological functions, such as mastication and speech. The tooth root is essential for these functions because it anchors the tooth to the jaw bone. During mastication, the root transmits and balances occlusal forces to the jaw bone through the periodontal ligament (PDL). The neurovascular bundle, which supplies blood flow, nutrition, and sensation to our teeth, also runs through the tooth root (*Li et al., 2017*). The loss of a functional root therefore reduces bone support to the tooth and adversely affects function of the dentition. Understanding how this organ patterning is determined will also provide information about organismic fitness through evolution (*Irvine and Harvey, 2015*). Teeth are the most common primate fossil remains due to their resistance to degradation; thus, morphological analysis of the tooth root can also provide critical clues about hominid evolution (*Emonet et al., 2014*). For example, a reduction in the number of premolar roots can be observed already in early hominins (*Emonet and Kullmer, 2014*). Neanderthal molars show elongated root trunks and apically positioned root furcations (*Kupczik and Hublin, 2010*; *Macchiarelli et al., 2006*). Therefore, investigating root development offers unique insights into organogenesis and human evolution.

**eLife digest** Different teeth have different numbers of roots. Incisors and canines each have one, and molars have two or three. Roots anchor the teeth to the jawbone, and provide a route for blood and nerves to reach the tooth. Getting the shape and number of the roots right during development is important to make sure that each tooth has proper support and function.

A protein called Ezh2 helps the bones of the face to develop, but it was not known how it affects how the roots of teeth grow. Teeth form from two layers of tissue; epithelium on the outside and mesenchyme on the inside. Jing et al. have now looked at what happens when Ezh2 is not present in these tissues in the molar teeth of developing mice.

The teeth of the mice were affected in different ways depending on which tissue Ezh2 was missing from. When the mesenchyme lacked Ezh2, the roots of the molar teeth did not form properly: the teeth formed too few roots, and the 'bridge' region between the roots did not develop correctly. When the epithelium lacked Ezh2, molar teeth formed the correct number of roots but the bridges between the roots developed later than normal. This suggests that signals from the mesenchyme determine how many roots each tooth grows.

Further investigation revealed that Ezh2 opposes the activity of another protein called Arid1a, and together they regulate the production of a protein that influences when cells divide. A balanced interaction between Ezh2 and Arid1a is important for tooth root development. This controls how bridges form between tooth roots, and ultimately determines how many roots a tooth grows.

Neanderthal teeth show evidence of forming bridges between roots later than modern human teeth, suggesting that similar regulation mechanisms have been important throughout human evolution. In the future, understanding how the roots of teeth form could help researchers to develop ways to regenerate teeth.

DOI: https://doi.org/10.7554/eLife.46426.002

Epithelial-mesenchymal interactions are required for root development and integration with the jawbone. The tooth root begins to develop with the guidance of a bilayered structure called Hertwig's epithelial root sheath (HERS). The cranial neural crest cell (CNCC)-derived mesenchyme forms the dental papilla and dental follicle. The mesenchyme of the apical papilla interacts with the inner layer of HERS and differentiates into odontoblasts that form dentin. The dental follicle also interacts with HERS and eventually produces the cementum, PDL, and adjacent alveolar bone (*Li et al., 2017*). Disruption of the interaction between HERS and the dental papilla or dental follicle leads to root development defects. For instance, if there is a disturbance to the developing HERS, differentiation of root odontoblasts will be compromised (*Kim et al., 2013*). HERS has been considered critical for determination of the tooth root number. It develops tongue-shaped epithelial protrusions (known as the epithelial diaphragm) that join horizontally to form a bridge, called the furcation, which constitutes the base of the pulp cavity and divides the roots. After the furcation forms, the apical growth of HERS drives root development in multi-rooted teeth, just as it does in single-rooted teeth. The different orientations of HERS in different types of teeth contribute to the formation of two-rooted lower molars, three-rooted upper molars, and single-rooted incisors (*Li et al., 2017*). However, the mechanisms involved in HERS regulation of furcation development remain unknown. Although previous studies have shown that changes in cell proliferation activity in the dental mesenchyme can lead to furcation defects (*Fons Romero et al., 2017*; *Sohn et al., 2014*), it is not clear whether the instructions that ultimately determine root furcation development and number reside in the dental mesenchyme or epithelium.

Recently, multiple signaling pathways have been implicated in the processes of root initiation and elongation (*Alfaqeeh et al., 2015*; *Kim et al., 2015*; *Li et al., 2015*; *Ono et al., 2016*), but how the number of tooth roots is determined remains unknown. Ezh2 is the enzymatic subunit of Polycomb repressive complex 2 (PRC2), a complex that methylates lysine 27 of histone H3 (H3K27) to promote transcriptional silencing. Polycomb proteins are an evolutionarily conserved family of chromatin regulators that serve to establish and maintain epigenetic memory during development (*Margueron and Reinberg, 2011*). Previous reports have indicated that craniofacial bone and cartilage formation are not detectable after loss of Ezh2 in neural crest cells, indicating a critical role for

Ezh2 in the determination of the osteochondrogenic lineage during craniofacial development (*Schwarz et al., 2014*).

The function of Ezh2 has recently been reported to be antagonized by other epigenetic factors. SWI/SNF chromatin remodeling complexes remodel nucleosomes and modulate gene transcription. In *Drosophila*, antagonism between polycomb and SWI/SNF complexes has been shown to regulate gene expression during development (*Kennison and Tamkun, 1988*). In humans, PRC2 and SWI/SNF complexes also antagonize each other during tumor formation (*Kadoch et al., 2016*; *Kadoch et al., 2017*). For example, Ezh2 inhibition leads to regression of ovarian tumors with mutations in Arid1a (a subunit of the SWI/SNF complex) (*Bitler et al., 2015*). However, whether and how these two opposing epigenetic regulating complexes regulate developmental patterning and morphogenesis in mammals still remains unknown.

In this study, we found that loss of Ezh2 in the tooth mesenchyme dramatically affects root patterning by transforming multi-rooted mouse molars into single-rooted ones, indicating a critical role for Ezh2 in determination of the molar root number via regulation of furcation development. In contrast, root furcation development was delayed after loss of Ezh2 in the epithelium and was unaffected after loss of Ezh2 in odontoblasts, suggesting that regulation of furcation development is determined through a mesenchymal signal. Significantly, Ezh2 and Arid1a work antagonistically to control Cdkn2a expression to coordinate furcation development and determine the root number. Our results have shown for the first time that the antagonistic interaction between Ezh2 and Arid1a plays a key role in regulating organogenesis in mammals. These findings provide a new understanding of the mechanism governing molar root number determination and may lead to applications for tooth regeneration in the future. Given that Neanderthal molars have long root trunks and delayed furcation formation in comparison to those of modern humans, our study highlights the significance of epigenetic regulation for the patterning of organs during human evolution.

## Results

### Ezh2 in the dental mesenchyme plays a key role in root patterning and furcation formation during molar root development

Ezh2 is a key enzyme of the PRC2 complex that is responsible for trimethylation of histone 3 lysine 27 (H3K27Me3). In order to investigate the role of Ezh2 in epigenetically regulating root patterning during tooth morphogenesis, we first analyzed the expression pattern of Ezh2 in developing molars. We found that Ezh2 is widely expressed in the dental epithelium, dental follicle, and dental papilla of control mice prior to root development initiation at the newborn stage (*Figure 1C*). H3K27Me3 was detectable in a similar pattern to that of Ezh2 in control mice, consistent with Ezh2's execution of a PRC2-dependent function during molar development (*Figure 1E*). In order to test the functional significance of Ezh2-mediated root patterning and development, we generated *Osr2-Cre;Ezh2^{fl/fl}* mice, in which *Ezh2* is specifically ablated in the dental mesenchyme. *Osr2-Cre* genetically targets the dental mesenchyme and alveolar bone but not the tooth epithelium; thus, we expected *Ezh2* expression to be lost from the mesenchyme of *Osr2-Cre;Ezh2^{fl/fl}* teeth, but to persist in the epithelium. Indeed, Ezh2 and H3K27Me3 were undetectable in the molar mesenchyme of *Osr2-Cre;Ezh2^{fl/fl}* mice at the newborn stage (*Figure 1D and F*), indicating efficient tissue-specific deletion of *Ezh2* in the dental mesenchyme.

There were no morphological differences between the crowns of *Osr2-Cre;Ezh2^{fl/fl}* and control molars at the newborn stage (*Figure 1A–1B*), prior to root development. At one week after birth, tooth crown formation is almost complete and root formation is yet to start. Tooth crown formation was similar in *Osr2-Cre;Ezh2^{fl/fl}* and control mice at one week of age (*Figure 2—figure supplement 1A–1F*), indicating that Ezh2 is dispensable for crown patterning.

In control mice, at two weeks after birth the root furcation was well formed, resulting in two roots in the mandibular first molars (*Figure 2A–2E*). Interestingly, only one root trunk with no furcation was observed in *Osr2-Cre;Ezh2^{fl/fl}* mandibular first molars (*Figure 2F–2J*). The absence of furcation persisted in *Osr2-Cre;Ezh2^{fl/fl}* mice at postnatal 4 weeks (*Figure 2P–2T*), by which time the tooth root had completed development in the control group (*Figure 2K–2O*). Moreover, the alveolar bone underneath the molar was undetectable throughout all developmental stages in *Osr2-Cre; Ezh2^{fl/fl}* mice. Interestingly, *Dspp* expression was not affected in *Osr2-Cre;Ezh2^{fl/fl}* mice, indicating

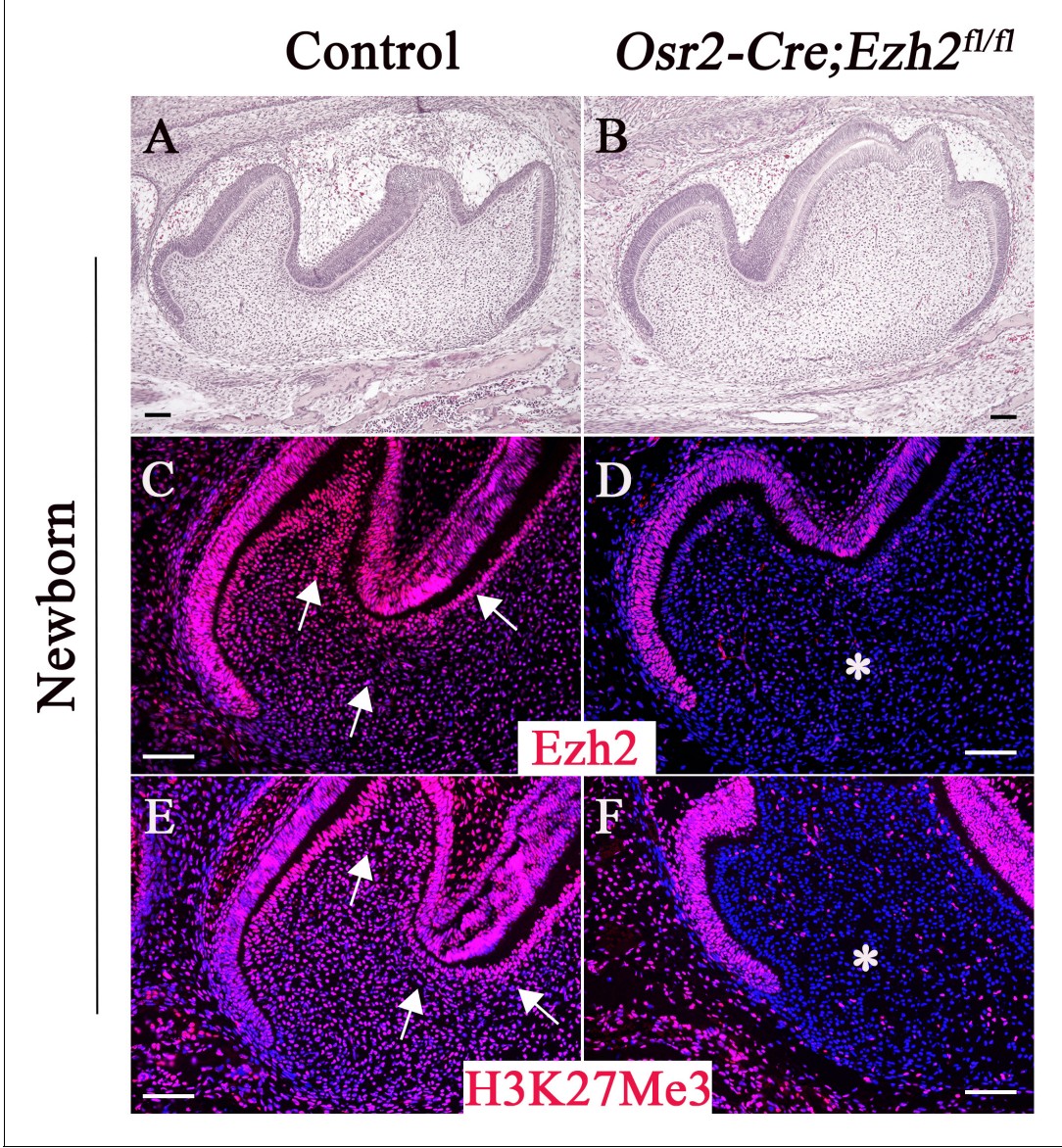

**Figure 1.** Loss of Ezh2 in the dental mesenchyme results in decreased H3K27Me3 histone methylation. H&E staining (**A–B**), Ezh2 immunofluorescence (**C–D**), and H3K27Me3 immunofluorescence (**E–F**) of newborn control and *Osr2-Cre;Ezh2*<sup>fl/fl</sup> molars. Arrows indicate positive signal and asterisks indicate absence of signal. n ≥ 3 histological sections were examined from multiple littermate mice per group. Scale bars, 100 µm.
DOI: https://doi.org/10.7554/eLife.46426.003

that loss of *Ezh2* in the dental mesenchyme has no effect on odontoblast differentiation (*Figure 2— figure supplement 2*).

In order to investigate whether mandibular and maxillary tooth furcations develop similarly, we also analyzed maxillary molars from two-week-old mice, which have three roots rather than the two of mandibular molars in controls (*Figure 2—figure supplement 1G–1J*). We found that maxillary molars were also single-rooted with no furcation formation in *Osr2-Cre;Ezh2*<sup>fl/fl</sup> mice (*Figure 2—figure supplement 1K–1N*), suggesting that the mechanisms regulating root patterning and furcation development are similar for maxillary and mandibular molars.

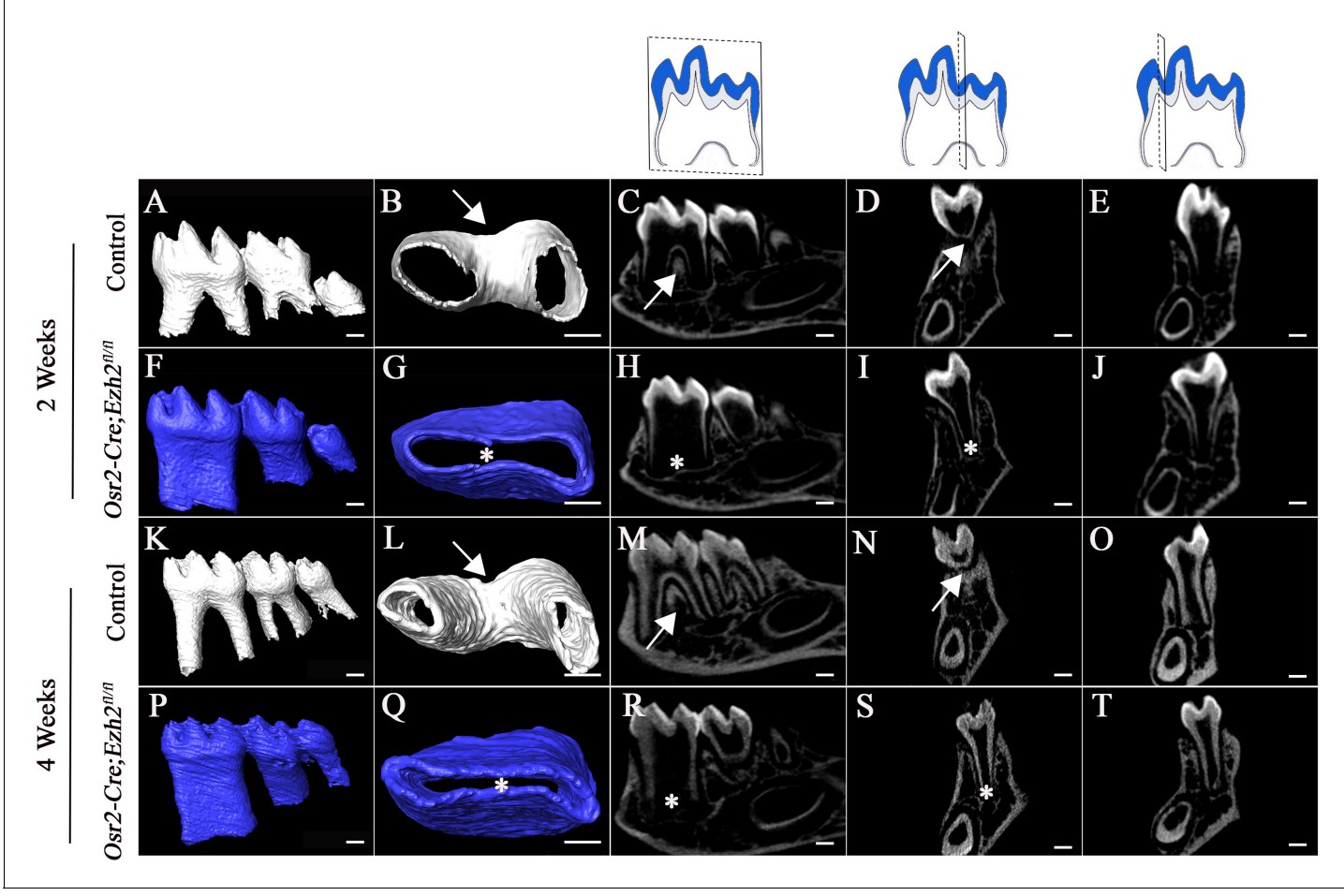

**Figure 2.** Loss of Ezh2 in the dental mesenchyme leads to single-rooted molars. MicroCT images of control (white) and *Osr2-Cre;Ezh2^{fl/fl}* (blue) molars at postnatal (PN) 2 and 4 weeks of age. A, F, K, P, lateral view of mandibular molars; B, G, L, Q, apical view of mandibular first molars; C, H, M, R, sagittal sections of mandibular molars; D, I, N, S, coronal sections of mandibular molars in the furcation region; E, J, O, T, coronal sections of mandibular molars in the root forming region. The schematic drawings indicate where the CT section were taken. Arrows indicate furcation and asterisks indicate absence of furcation. Scale bars, 200 µm.

DOI: https://doi.org/10.7554/eLife.46426.004

The following figure supplements are available for figure 2:

**Figure supplement 1.** Loss of Ezh2 in the dental mesenchyme leads to single-rooted molars in the upper jaw.

DOI: https://doi.org/10.7554/eLife.46426.005

**Figure supplement 2.** Loss of Ezh2 in the dental mesenchyme has no effect on odontoblast differentiation.

DOI: https://doi.org/10.7554/eLife.46426.006

## Loss of Ezh2 in the dental mesenchyme affects epithelial diaphragm, alveolar bone, and PDL formation

At the beginning of root formation, the root sheath forms the epithelial diaphragm. Previous studies highlighted the importance of differential growth of the epithelial diaphragm as the crucial step in forming multi-rooted molars (*Li et al., 2017*). In order to test whether formation of the epithelial diaphragm was affected in *Osr2-Cre;Ezh2^{fl/fl}* mice, we investigated its development at earlier time points. At PN 1 week, the epithelial diaphragm was not fused at the furcation region in control mice, as evidenced by a lack of continuous Krt14 staining in the apical region of the molar (*Figure 3A–3B*). One day later, a fused epithelial diaphragm was detectable in control mice (*Figure 3G–3H*). In contrast, we did not detect epithelial diaphragms in *Osr2-Cre;Ezh2^{fl/fl}* mice at any time point (*Figure 3D–3E and J–K*). Moreover, cell proliferation activity in the mesenchyme of the apical region was compromised in molars of *Osr2-Cre;Ezh2^{fl/fl}* mice just prior to the time point at which epithelial

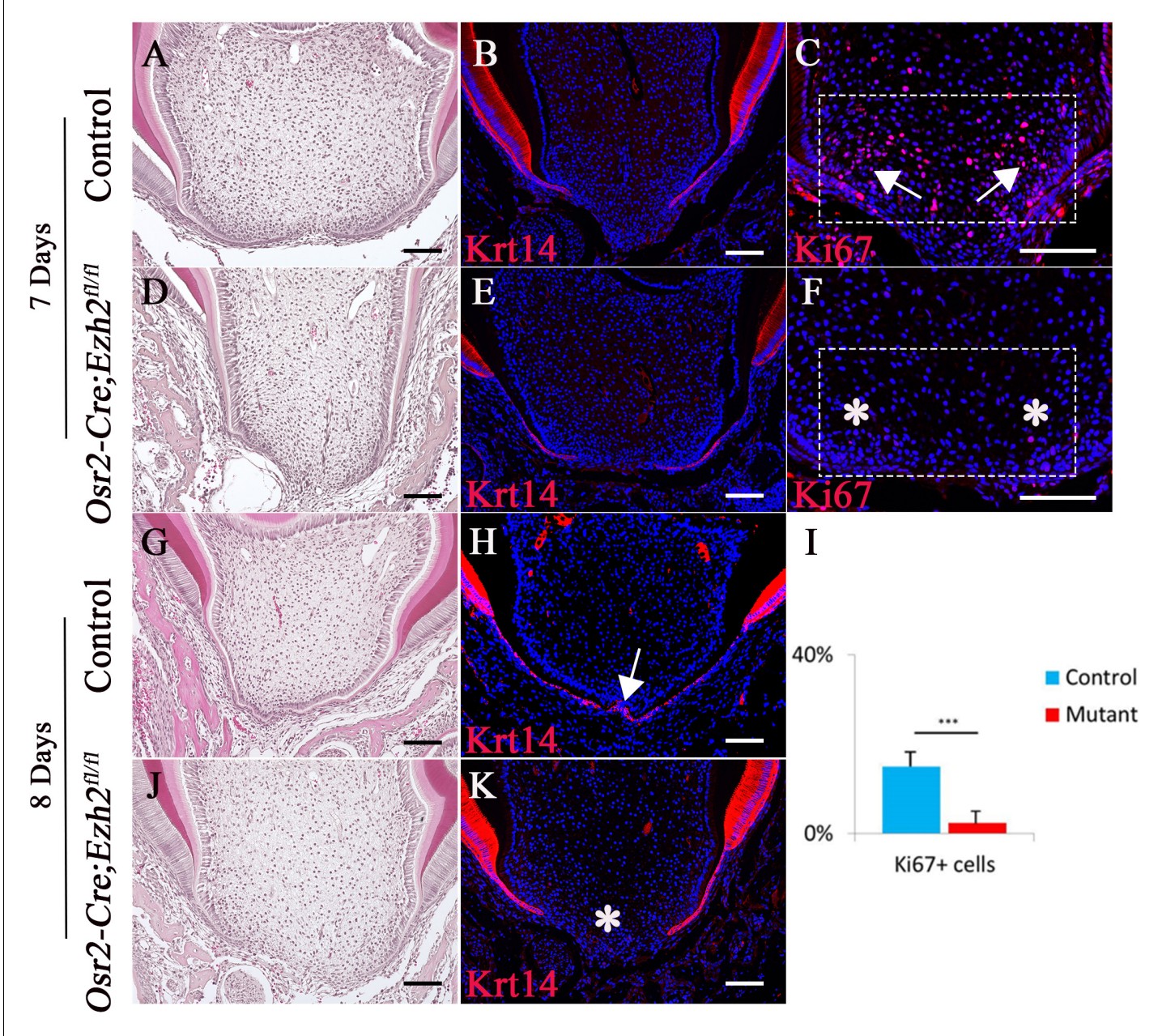

**Figure 3.** Loss of Ezh2 in the dental mesenchyme affects epithelial diaphragm formation. A-K, H&E staining (**A, D, G, J**), Krt14 immunofluorescence (**B, E, H, K**) and Ki67 staining (**C, F**) of coronal sections of control and *Osr2-Cre;Ezh2*fl/fl (mutant) molars at postnatal 7 days and 8 days. I, Quantitation of Ki67+ cells in boxed areas of C and F presented as mean ± SD with n = 4. Arrows indicate positive signal and asterisks indicate absence of signal. ***, p<0.001. Scale bars, 100 μm. Statistical analyses were performed using two-tailed Student's t-test.

DOI: https://doi.org/10.7554/eLife.46426.007

The following source data and figure supplement are available for figure 3:

**Source data 1.** Source data for *Figure 3I*.
DOI: https://doi.org/10.7554/eLife.46426.009
**Figure supplement 1.** Loss of Ezh2 in the dental mesenchyme has no effect on cell survival.
DOI: https://doi.org/10.7554/eLife.46426.008

diaphragm formation would normally be expected (*Figure 3C, F and I*). However, no apoptotic cells were detectable in the molars of control or *Osr2-Cre;Ezh2^{fl/fl}* mice at PN 1 week or PN 3 weeks of age, indicating that loss of *Ezh2* in the tooth mesenchyme has no impact on cell survival (*Figure 3— figure supplement 1*).

In addition, formation of the alveolar bone and PDL was abnormal in *Osr2-Cre;Ezh2^{fl/fl}* mice. At PN 2 weeks, alveolar bone was already formed between and underneath the molars of control mice. However, alveolar bone was undetectable in *Osr2-Cre;Ezh2^{fl/fl}* mice until PN 4 weeks (*Figure 4A– 4B, D–E and G–J*). Similarly, expression of PDL marker periostin was detectable in two-week-old control mice, but its expression was undetectable in *Osr2-Cre;Ezh2^{fl/fl}* mice (*Figure 4C and F*), indicating defective PDL formation due to loss of *Ezh2* in the dental follicle. Collectively, our studies show that Ezh2 in the dental mesenchyme is crucial for the development of alveolar bone and PDL.

## Ezh2 in the dental epithelium and odontoblasts is not required for root patterning or furcation formation

*Ezh2* is also expressed in the epithelium of the mouse molar. In order to investigate whether Ezh2 in the epithelium is crucial for root patterning and furcation development, we generated *Krt14-Cre; Ezh2^{fl/fl}* mice. At PN 2 weeks, the furcation was already formed in control molars, but it was not detectable in molars of *Krt14-Cre;Ezh2^{fl/fl}* mice (*Figure 5A–5J*). Interestingly, alveolar bone formation was delayed in *Krt14-Cre;Ezh2^{fl/fl}* mice at PN 2 weeks (*Figure 5—figure supplement 1A–1B*).

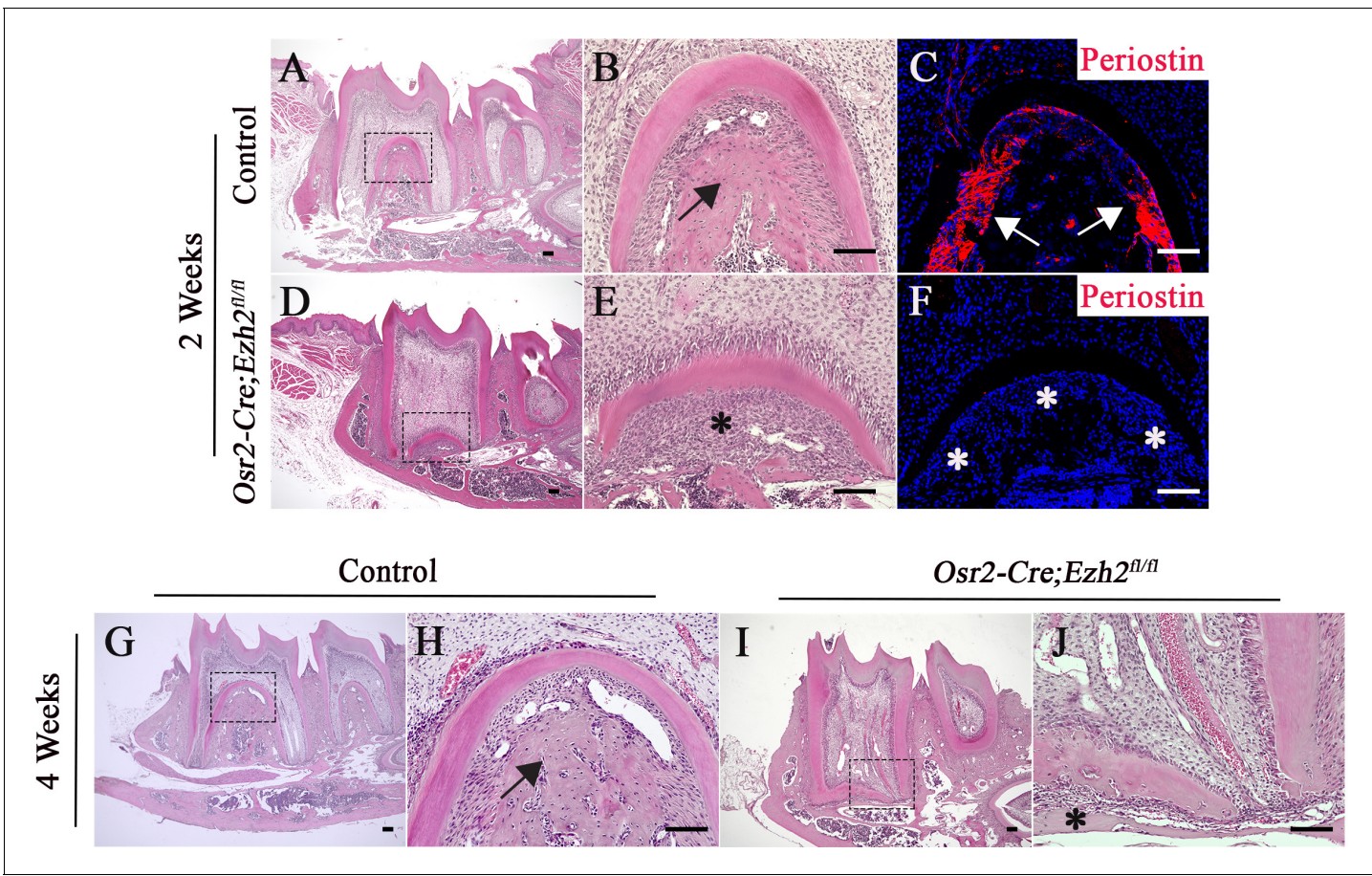

**Figure 4.** The alveolar bone and periodontal ligament are affected in *Osr2-Cre;Ezh2^{fl/fl}* molars. (A-F) H&E staining (A, B, D, E) and Periostin immunofluorescence (C, F) of control and *Osr2-Cre;Ezh2^{fl/fl}* molars at 2 weeks of age. B and E are magnified images of the boxed areas in A and D, respectively. (G-J) H&E staining of control and *Osr2-Cre;Ezh2^{fl/fl}* molars at 4 weeks of age. (H and J) are magnified images of the boxed areas in G and I, respectively. Arrows indicate normal alveolar bone and asterisks indicate defective alveolar bone. n ≥ 3 histological sections were examined from multiple littermate mice per group. Scale bars, 100 µm.
DOI: https://doi.org/10.7554/eLife.46426.010

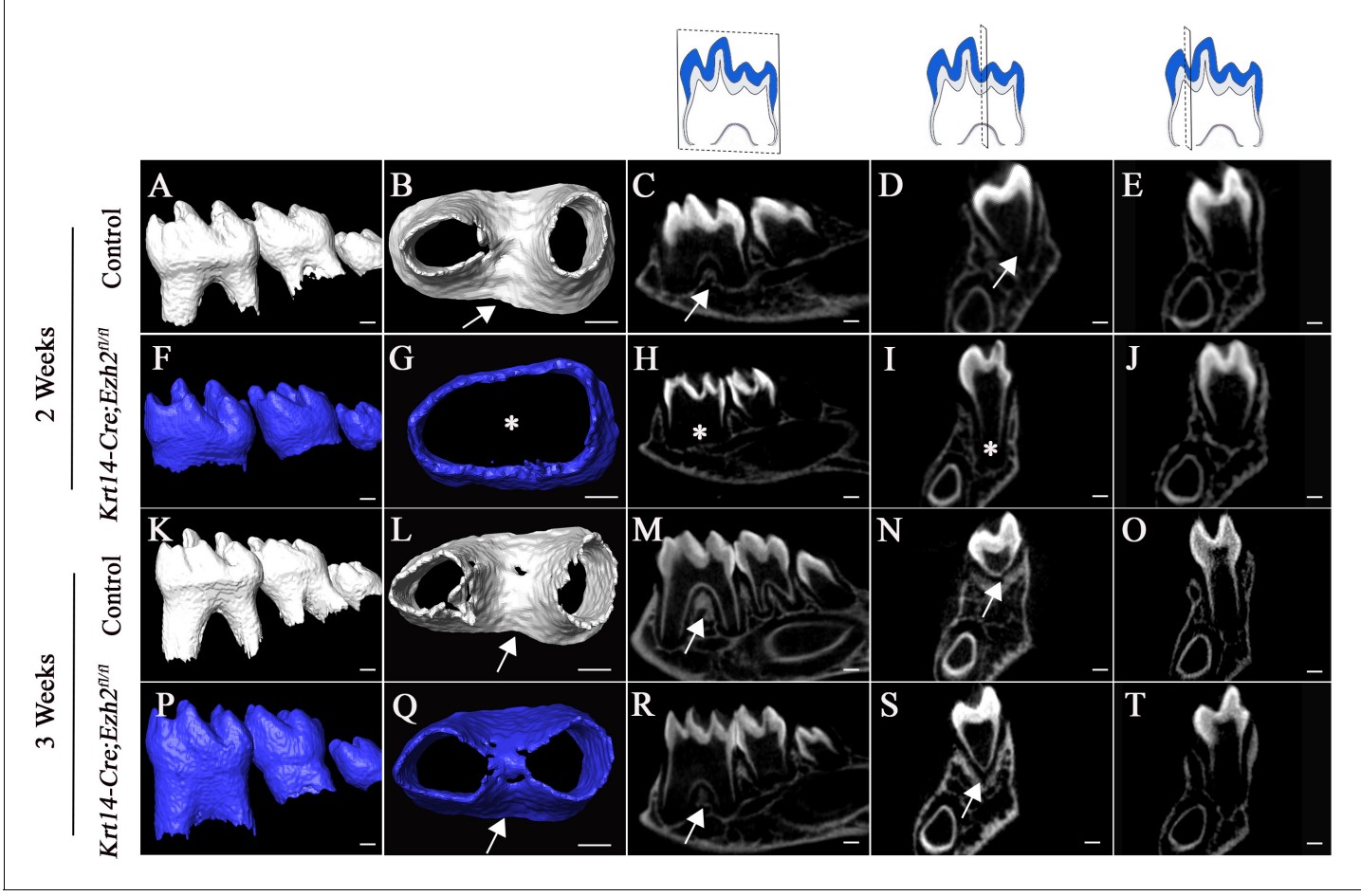

**Figure 5.** Loss of *Ezh2* in the epithelium leads to delayed furcation development. MicroCT images of control (white) and *Krt14-Cre;Ezh2^{fl/fl}* (blue) molars at 2 and 3 weeks of age. A, F, K, P, lateral view of mandibular molars; B, G, L, Q, apical view of mandibular first molars; C, H, M, R, sagittal sections of mandibular molars; D, I, N, S, coronal sections of mandibular molars in the furcation region; E, J, O, T, coronal sections of mandibular molars in the root forming region. The schematic drawings indicate where the CT section were taken. Arrows indicate furcation and asterisks indicate absence of furcation. Scale bars, 200 μm.

DOI: https://doi.org/10.7554/eLife.46426.011

The following figure supplements are available for figure 5:

**Figure supplement 1.** H&E staining of molars in control and *Krt14-Cre;Ezh2^{fl/fl}* mice.

DOI: https://doi.org/10.7554/eLife.46426.012

**Figure supplement 2.** Root furcation development is unaffected after loss of Ezh2 in odontoblasts.

DOI: https://doi.org/10.7554/eLife.46426.013

**Figure supplement 3.** Arid1a expression is not affected in *Osr2-Cre;Ezh2^{fl/fl}* molars.

DOI: https://doi.org/10.7554/eLife.46426.014

However, by PN 3 weeks, the alveolar bone and molar root furcation had formed in *Krt14-Cre;Ezh2^{fl/fl}* mice (*Figure 5K–5T* and *Figure 5—figure supplement 1C–1F*), indicating delayed furcation development due to loss of Ezh2 in the dental epithelium.

Next, we investigated whether Ezh2 in odontoblasts has a role in root patterning and furcation development by generating *Dmp1-Cre;Ezh2^{fl/fl}* mice. We found no distinguishable differences between the tooth roots of *Dmp1-Cre;Ezh2^{fl/fl}* and control mice at PN 3 weeks, based on CT images and H&E staining (*Figure 5—figure supplement 2*). Alveolar bone and PDL formation were also normal in *Dmp1-Cre;Ezh2^{fl/fl}* mice when compared to control samples. Collectively, our results indicate that Ezh2 in odontoblasts is not required for root patterning and furcation development.

## Antagonistic interaction between Ezh2 and Arid1a in regulating root patterning

Arid1a is part of the SWI/SNF chromatin remodeling complex and has an antagonistic relationship with Ezh2 of the PRC2 complex in cancer development (*Bitler et al., 2015*; *Wu et al., 2018*). Arid1a has a similar expression pattern to that of Ezh2 during root development. Interestingly, it appears that the expression of Arid1a was not affected in the molars of *Osr2-Cre;Ezh2^{fl/fl}* mice (*Figure 5— figure supplement 3*), indicating Arid1a is not a downstream target of Ezh2. However, previous studies have shown that the antagonism between Ezh2 and Arid1a may occur on the functional level (*Bitler et al., 2015*). In order to investigate whether there is an antagonistic interaction between Ezh2 and Arid1a in regulating furcation development, we generated *Osr2-Cre;Ezh2^{fl/fl};Arid1a^{fl/+}* mice. Indeed, the abnormal root patterning and furcation development seen in *Osr2-Cre;Ezh2^{fl/fl}* mice were completely rescued in *Osr2-Cre;Ezh2^{fl/fl};Arid1a^{fl/+}* molars, based on microCT images

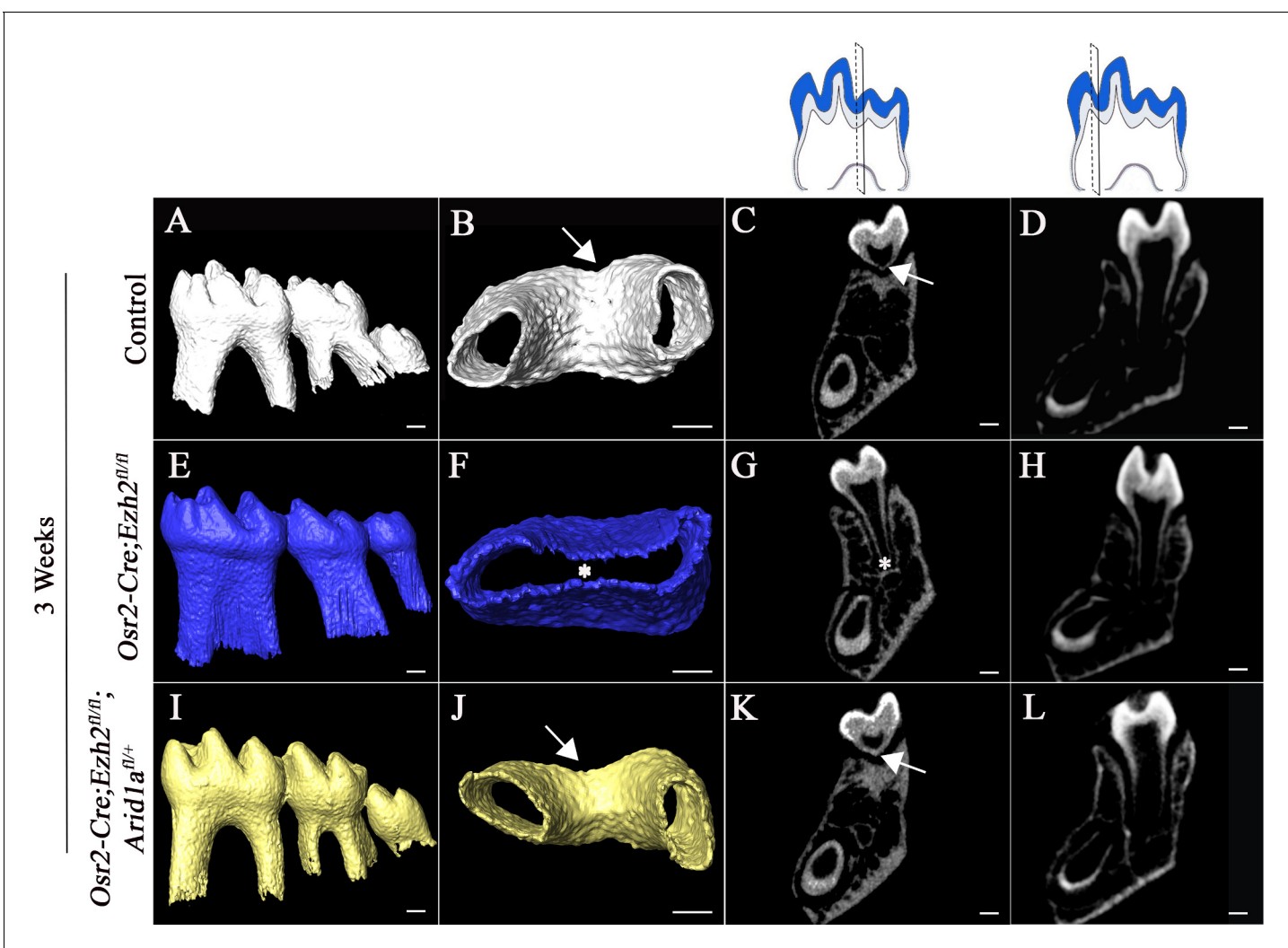

**Figure 6.** Furcation development is rescued in *Osr2-Cre;Ezh2^{fl/fl};Arid1a^{fl/+}* mice. MicroCT images of control, *Osr2-Cre;Ezh2^{fl/fl}* and *Osr2-Cre;Ezh2^{fl/fl}; Arid1a^{fl/+}* molars at 3 weeks of age. A, E, I, lateral view of mandibular molars; B, F, J, apical view of mandibular first molars; C, G, K, coronal sections of molars in the furcation region; D, H, L, coronal sections of mandibular molars in the root forming region. The schematic drawings indicate where the CT section were taken. Arrows indicate furcation and asterisks indicate absence of furcation. Scale bars, 200 μm.

DOI: https://doi.org/10.7554/eLife.46426.015

The following figure supplement is available for figure 6:

**Figure supplement 1.** Monoallelic deletion of Arid1a has no effect on furcation formation.
DOI: https://doi.org/10.7554/eLife.46426.016

(*Figure 6A–6L*), indicating that Arid1a and Ezh2 may work antagonistically to control furcation development. To examine whether monoallelic deletion of *Arid1a* affects furcation development, we generated *Osr2-Cre;Arid1a^fl/+^* mice and found that their root furcations were identical to those of control mice (n = 4), suggesting that haploinsufficiency of Arid1a did not affect root patterning or development (*Figure 6—figure supplement 1*). We also analyzed alveolar bone and PDL development at PN 3 weeks and found that there was no difference between the molars of *Osr2-Cre;Ezh2^fl/fl^;Arid1a^fl/+^* and control mice, suggesting that alveolar bone and PDL devlopment was also rescued in *Osr2-Cre;Ezh2^fl/fl^;Arid1a^fl/+^* mice (*Figure 6*).

In order to identify downstream mediators that control furcation development, mesenchymal tissue from PN day three mouse molars was isolated for RNA-seq analysis. We found that more genes were upregulated than downregulated in *Osr2-Cre;Ezh2^fl/fl^* molars (*Figure 7A*), consistent with the gene repression function of Ezh2. Patterning genes such as Hox family members and *Hand2* were highly enriched and among the top twenty upregulated genes, indicating their potential role in root furcation development. The proliferation of dental mesenchymal cells has been shown to regulate tooth root furcation formation (*Sohn et al., 2014*). Interestingly, we found that *Cdkn2a*, a cell cycle inhibitor, was also upregulated in the root-forming dental mesenchyme in *Osr2-Cre;Ezh2^fl/fl^* molars, consistent with the observed reduction in cell proliferation activity. Therefore, we hypothesized that Cdkn2a may be a downstream target of Ezh2 involved in root furcation development. In order to test our hypothesis, we first examined the expression of Cdkn2a in the molar root-forming region and found that it was upregulated in *Osr2-Cre;Ezh2^fl/fl^* mice (*Figure 7B and D*), whereas the level of Cdkn2a expression was restored to the level in control samples in *Osr2-Cre;Ezh2^fl/fl^;Arid1a^fl/+^* molars (*Figure 7F*). Based on this finding, we further examined cell proliferation activity in the root-forming region and found that proliferation was also restored in *Osr2-Cre;Ezh2^fl/fl^;Arid1a^fl/+^* molars (*Figure 7C, E, G and H*). Furthermore we performed the CHIP sequencing of H3K27Me3 in root mesenchyme of the control molars. Interestingly, our data have shown that the Hox genes and Cdkn2a are in the H3K27Me3 binding sites (*Figure 7—figure supplement 1*), which is consistent with our RNA sequencing data. Collectively, our data highlight a critical role for the antagistic interaction between Ezh2 and Arid1a in controlling Cdkn2a expression in regulating cell proliferation during root patterning and furcation development.

## Discussion

### Mesenchymal signaling controls root patterning during tooth morphogenesis

Epithelial-mesenchymal interaction is crucial for organ patterning and morphogenesis. During the formation of branched organs, the mesenchyme can instruct the epithelium to form branching patterns. For example, various types of signaling in the mesenchyme, including WNT, hedgehog (HH) and bone morphogenetic protein (BMP), play important roles in regulating branch patterning and morphogenesis in the salivary gland, kidney and lung (*Lu and Werb, 2008*). Similarly, epithelial-mesenchymal interaction is also crucial for tooth root patterning and morphogenesis. Previous studies have suggested that the pattern of HERS growth may correlate with the number, length, and shape of roots (*Kumakami-Sakano et al., 2014*). Furthermore, HERS provides instructive signals that contribute to the induction of dental mesenchyme differentiation, suggesting that it functions as a signaling center to guide root formation (*Huang et al., 2009*; *Li et al., 2017*). For example, HERS-derived TGFβ/BMP signaling regulates root dentin formation through Nfic expression in the dental mesenchyme (*Huang et al., 2010*). Previous studies have suggested that the pattern of the cervical epithelial diaphragm may guide furcation formation, and signals from HERS may have a critical impact on determination of the root number. For instance, the Eda pathway is specifically active in HERS in mouse molars. Cell proliferation activity is altered in the dental mesenchyme in *Eda* mutant molars with delayed furcation formation, suggesting that epithelial-derived signals may regulate furcation development through epithelial-mesenchymal interaction (*Fons Romero et al., 2017*). Recent studies have begun to explore the role of the dental mesenchyme in regulating root patterning and furcation development. The directionality of HERS growth may be regulated by differential proliferation of mesenchymal cells in furcation-forming and root-forming regions, which in turn determines

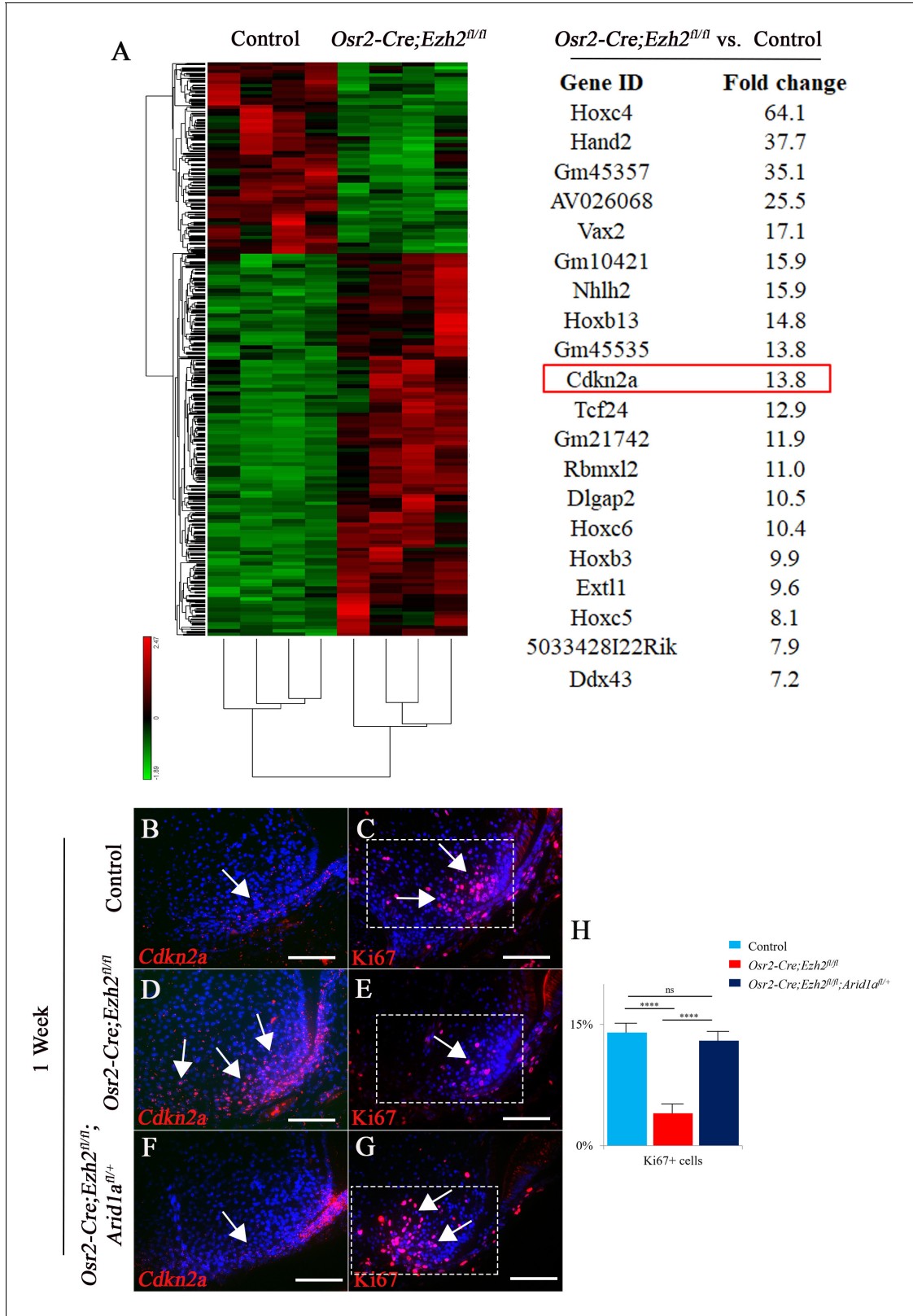

**Figure 7.** RNA-sequencing analysis from control and *Osr2-Cre;Ezh2*[fl/fl] molars. (**A**) Heatmap and list of top twenty upregulated genes (red highlights Cdkn2a) generated from RNA-sequencing. (**B-G**) *Cdkn2a* and Ki67 staining of control, *Osr2-Cre;Ezh2*[fl/fl] and *Osr2-Cre;Ezh2*[fl/fl];*Arid1a*[fl/+] molars at 1 week of age. (**H**) quantitation of Ki67+ cells in boxed areas of C, E and G presented as mean ± SD with n = 4. Arrows indicate positive signal. ns, not significant. ***, p<0.001. Scale bars, 100 μm. Statistical analyses were performed using one-way ANOVA.

*Figure 7 continued on next page*

*Figure 7 continued*

DOI: https://doi.org/10.7554/eLife.46426.017

The following source data and figure supplement are available for figure 7:

**Source data 1.** Source data for *Figure 7H*.

DOI: https://doi.org/10.7554/eLife.46426.019

**Figure supplement 1.** ChIP-seq signals of H3K27Me3 ChIP-sequencing from wild type mouse molars.

DOI: https://doi.org/10.7554/eLife.46426.018

root number (*Sohn et al., 2014*). However, the key determinant for root patterning remained unknown.

In this study, we found that loss of *Ezh2* in the tooth mesenchyme transformed multi-rooted molars into single-rooted ones in the mouse, suggesting the significance of the dental mesenchyme in regulating root pattern and furcation development. In contrast, loss of *Ezh2* in the dental epithelium resulted in delayed furcation development without affecting root patterning. These data suggest that signals from the mesenchyme, rather than the epithelium, are the driving force behind tooth patterning. It is possible that loss of Ezh2 in the dental mesenchyme affects the growth of HERS through mesenchymal-epithelial interaction. This is highlighted by the phenotype in which the epithelial diaphragm fails to fuse at the future furcation site in *Osr2-Cre;Ezh2^{fl/fl}* mice. Previous studies have also highlighted that root defects are mainly related to signaling alterations in the mesenchyme rather than the epithelium (*Li et al., 2017*). Moreover, in humans, mutations adversely affecting the dental mesenchyme are closely associated with tooth root defects. For example, dentinogenesis imperfecta type I (attributed to mutations in *COL1A1* and *COL1A2*) and dentinogenesis imperfecta type II (caused by mutations in *DSPP*) both involve root defects. Furthermore, X-linked hypophosphatemia (linked to mutations in *PHEX*), which results in hypomineralized dentin and enlarged pulp cavities, results in a phenotype similar to taurodontism without apical displacement of the furcation (*Fong et al., 2009*; *Li et al., 2017*). Taking all these lines of evidence together, we conclude that mesenchyme-derived signals are the key determinant of root patterning.

It is interesting to note that dental cusp and root patterning can be regulated independently because loss of Ezh2 in the dental mesenchyme does not adversely affect dental cusp patterning though it results in a root patterning defect. Importantly, alveolar bone formation is intimately linked to root patterning and development. It is well known that proper integration between the dental root and alveolar bone is of paramount importance for our dentition. Future study will allow us to investigate whether there are common or predetermined progenitor cells that contribute both to root and alveolar bone formation. Osr2-Cre is active in the dental mesenchyme including both the dental follicle and dental papilla. The dental follicle gives rise to periodontal tissues such as alveolar bone and PDL, which have a critical impact on tooth root development and tooth eruption (*Takahashi et al., 2019*), implying the interaction between periodontal tissue and tooth root development. In our study, loss of alveolar bone and PDL in the mouse molar correlated with the observed root furcation defect. However, whether the alveolar bone and PDL defects are primary malformations or the result of root furcation defect still needs to be further investigated.

## Epigenetic factors in regulating organ patterning and development

Although various signaling pathways have been reported to play crucial roles in organ patterning and morphogenesis, such as BMP, TGFβ, WNT, FGF, and HH, the function of epigenetic regulation in organ patterning is largely unknown. The antagonism between PRCs and SWI/SNF complexes is crucial in both development and disease. For example, SWI/SNF antagonizes Polycomb-mediated transcriptional repression and suppresses Cyclin E transcription, arresting the cell division of myogenic precursors during muscle differentiation (*Ruijtenberg and van den Heuvel, 2015*). In human malignant rhabdoid tumors, loss of SMARCB1 (a subunit of SWI/SNF) leads to Polycomb-mediated repression of genes that suppress proliferation; when SMARCB1 is re-expressed, Polycomb is removed from the chromatin and DNA methylation is lost (*Kadoch et al., 2016*). However, there is virtually no information on how PRCs and SWI/SNF exert epigenetic control over organ patterning and development in mammals. In this study, we found that antagonistic interaction between Ezh2 and Arid1a is indispensable for tooth root furcation patterning. Interestingly, Ezh2 represses the Hox

gene family, and many Hox genes can suppress osteochondrogenesis (*Creuzet et al., 2002*). In particular, cells in pharyngeal arch one and the anterior domains of neural crest cells (NCCs) do not express Hox genes, thus enabling the cartilage and bony elements of the face to form (*Minoux et al., 2017*). Although migration of NCCs and their localization to target structures are not impaired by loss of Ezh2, craniofacial osteochondrogenesis is suppressed in *Wnt1-Cre;Ezh2^{fl/fl}* mice (*Schwarz et al., 2014*). In our study, we found that loss of *Ezh2* in the dental mesenchyme also affected dental follicle-derived tooth root-supporting tissue including PDL and alveolar bone, likely through overactivation of Hox genes, indicating that the differentiation of dental follicle-derived cells is also Ezh2-dependent.

A previous study has shown that inhibition of Ezh2 methyltransferase activity can inhibit tumor cells in an *Arid1a*-mutated model, highlighting the antagonism of Ezh2 and Arid1a in tumor formation. Interestingly, Arid1a knockdown does not affect Ezh2 expression, and the antagonism between Ezh2 and Arid1a occurs on the functional level (*Bitler et al., 2015*). Similarly, in our study, we found that loss of *Ezh2* does not affect Arid1a expression but instead works antagonistically with Ezh2 to control the furcation pattern, possibly via regulation of Cdkn2a, suggesting Ezh2 may antagonize Arid1a on the functional level in this domain as well. Cdkn2a is a well-known cell cycle inhibitor. Previous studies have shown that Cdkn2a is involved in cell cycle regulation in various physiological process as a downstream target of PRC2. For example, Cdkn2a serves as a cell cycle regulator downstream of Ezh2 in a variety of cancers (*Kim and Roberts, 2016*). In our study, Cdkn2a is likely the master cell cycle regulator that controls the root furcation development via regulating the cell proliferation activity in the root apical region. Taken together, our results highlight the importance of the fine-tuned balance between antagonistic epigenetic regulators Ezh2 and Arid1a in tooth root patterning and development.

From an evolutionary perspective, our results clearly demonstrate that epigenetic regulation plays a key role in dental root patterning and development. Neanderthal molars have a taurodont phenotype with a longer root trunk than the ones seen in anatomically modern humans and show late bifurcation or trifurcation of the roots (*Macchiarelli et al., 2006*). In our study, loss of *Ezh2* in the tooth epithelium recaptures the taurodont phenotype, indicating the potential role of epithelial Ezh2 in human evolution. Importantly, it has been suggested that the root attachment area is adaptively linked to the differing occlusal loads and mechanical resistance levels of foods eaten by mammals. For instance, primates that eat hard substances exhibit larger root surface areas than those that feed on less mechanically resistant foods (*Kupczik and Hublin, 2010*). Some of the ways by which selective mechanisms may have operated to maximize root surface area are to increase the number of roots or lengthen the root, thus stabilizing the dentition. The well-separated dental roots offer improved stability for our dentition within the jawbones in modern humans. Collectively, a better understanding of the mechanisms involved in determination of tooth root patterning and development can therefore provide critical clues about human evolution, as well as potential therapeutic approaches to tooth regeneration.

# Materials and methods

**Key resources table**

| Reagent type (species) or resource | Designation | Source or reference | Identifiers | Additional information |
|---|---|---|---|---|
| Strain, strain background (M. musculus) | *Arid1a^{flox/flox}* | Jackson Laboratory | Stock No. 027717; RRID:IMSR_JAX:027717 | |
| Strain, strain background (M. musculus) | *Dmp1-Cre* | Jackson Laboratory | Stock No. 023047; RRID:IMSR_JAX:023047 | |
| Strain, strain background (M. musculus) | *Ezh2^{flox/flox}* | Jackson Laboratory | Stock No. 022616; RRID:IMSR_JAX:022616 | |

*Continued on next page*

*Continued*

| Reagent type (species) or resource | Designation | Source or reference | Identifiers | Additional information |
|---|---|---|---|---|
| Strain, strain background (M. musculus) | *Krt14-Cre* | Jackson Laboratory | Stock No. 018964; RRID:IMSR_JAX:018964 | |
| Strain, strain background (M. musculus) | *Osr2-Cre* | Rulang Jiang, Cincinnati Children's Hospital | | |
| Genetic reagent (M. musculus) | anti-Cdkn2a probe | Advanced Cell Diagnostics | Cat# 411011 | |
| Antibody | Rabbit monoclonal anti-Arid1a | Abcam | Cat# ab182561 | (1:100) |
| Antibody | Rabbit monoclonal anti-Ki67 | Abcam | Cat# ab16667; RRID:AB_302459 | (1:200) |
| Antibody | Rabbit monoclonal anti-Ezh2 | Cell Signaling Technology | Cat# 5246S; RRID:AB_10694683 | (1:200) |
| Antibody | Rabbit monoclonal anti-H3K27Me3 | Cell Signaling Technology | Cat# 9733S; RRID:AB_2616029 | (1:100) |
| Antibody | Rabbit polyclonal anti-Periostin | Abcam | Cat# ab14041; RRID:AB_2299859 | (1:100) |
| Antibody | anti-Rabbit Alexa Fluor 568 | Life Technologies | Cat# A-11011; RRID:AB_143157 | (1:200) |
| Antibody | anti-Mouse Alexa Fluor 568 | Life Technologies | Cat# A-11004; RRID:AB_2534072 | (1:200) |
| Commercial assay or kit | In Situ Cell Death Detection Kit | Roche Life Science | Cat# 11684795910 | |
| Commercial assay or kit | RNeasy Micro Kit | QIAGEN | Cat# 74004 | |
| Commercial assay or kit | Chromatrap Enzymatic Shearing Kit | Chromatrap | Cat# 500165 | |
| Software, algorithm | ImageJ | NIH | RRID:SCR_003070 | |
| Software, algorithm | GraphPad Prism | GraphPad Software | RRID:SCR_002798 | |

## Animals and procedures

*Arid1a^(fl/fl)* (**Gao et al., 2008**), *Dmp1-Cre* (**Lu et al., 2007**), *Ezh2^(fl/fl)* (**Shen et al., 2008**), *Krt14-Cre* (**Fell et al., 2014**), and *Osr2-Cre* (gift from Rulang Jiang, Cincinnati Children's Hospital, **Tian et al., 2017**) mouse lines were used and cross-bred as needed in this study. All mouse experiments were conducted in accordance with protocols approved by the Department of Animal Resources and the Institutional Animal Care and Use Committee of the University of Southern California.

All mice were housed in pathogen-free conditions and analyzed in a mixed background. Mice were identified by ear tags. Genotyping was conducted on tail samples. Tail biopsies were lysed through incubation at 55°C overnight in DirectPCR tail solution (Viagen 102 T) followed by 85°C heat inactivation for 30 min and PCR-based genotyping (GoTaq Green MasterMix, Promega, and C1000 Touch Cycler, Bio-rad). Mice were euthanized by carbon dioxide overdose followed by cervical dislocation. All mice were used for analysis regardless of sex.

## Immunofluorescence and in situ hybridization (ISH)

For immunofluorescence analysis, mouse mandibles were dissected, fixed in 4% PFA overnight, and decalcified with 10% EDTA for 4 weeks. Then, the tissues were incubated with 15% sucrose for 2 hr and 30% sucrose overnight, followed by embedding in OCT. Frozen tissue blocks were sectioned at 10 mm on a cryostat (Leica) and mounted on SuperFrost Plus slides (Fisher). The tissue sections were blocked for 1 hr at room temperature in blocking solution (Vector Laboratories). Sections were then incubated with primary antibodies diluted in blocking solution at 4°C overnight. After washing three

times with PBS, sections were incubated with secondary antibodies in blocking solution at room temperature for 1 hr. DAPI was used for nuclear staining and all images were acquired using a Keyence microscope (Carl Zeiss).

In situ hybridization was performed using RNAscope multiplex fluorescent assay (Advanced Cell Diagnostics). Briefly, tissues were fixed in 4% PFA overnight at room temperature before cryosectioning. ISH was performed on 10 µm sections according to the manufacturer's instructions.

### MicroCT analysis

MicroCT analysis was performed using a SCANCO µCT50 device at the University of Southern California Molecular Imaging Center. The microCT images were acquired with the x-ray source at 70 kVp and 114 µA. The data were collected at a resolution of 10 µm. Three-dimensional (3D) reconstruction was done with AVIZO 7.1 (Visualization Sciences Group).

### TUNEL assays

Specimens were harvested, fixed overnight in 4% PFA, and decalcified in 10% EDTA for four weeks. Tissues were embedded in OCT compound (Sakura Tissue-Tek 4583), frozen, and sectioned at 8–10 µm thickness. Apoptotic cells were detected with the In Situ Cell Death Detection Kit (Roche Life Science 11684795910) following the recommended protocol.

### RNA-sequencing

Molar samples from three-day-old *Ezh2*$^{fl/fl}$ (control) and *Osr2-Cre;Ezh2*$^{fl/fl}$ mice (n = 4 per group) were collected for RNA isolation with RNeasy Micro Kit (QIAGEN). The quality of RNA samples was determined using an Agilent 2100 Bioanalyzer and all samples for sequencing had RNA integrity (RIN) numbers > 7.0. cDNA library preparation and sequencing were performed at the Epigenome Center of the University of Southern California. Single-end reads with 75 cycles were performed on Illumina Hiseq 4000 equipment for three pairs of samples. Raw reads were trimmed, aligned using TopHat (version 2.0.8) with the mm10 genome, and normalized using RPKM. Differential expression was calculated by selecting transcripts that changed with a significance of $p < 0.05$.

### ChIP-sequencing

Molar samples from three-day-old wildtype mice were collected to performed ChIP-sequencing using H3K27me3 antibody (Cell signaling) and Chromatrap Enzymatic Shearing Kit (Chromatrap). ChIP DNA was quantified by Bioanalyzer and sequencing libraries construction were prepared using the standard Illumina ChIP-seq protocol. Technology Center for Genomic and Bioinformatics, University of California, Los Angeles constructed the library and sequenced the ChIPseq libraries on Illumina Nextseq 500 platform. Reads were mapped to NCBI mouse reference genome (Genome Reference Consortium Mouse Build 38, Jan 2012) using Burrows-Wheeler Alignment (BWA) tool. The uniquely mapped reads were used to identify the regions in the genome with significant enrichment of H3K27me3 modification. The aligned bam files were sorted using SAMtools followed by peak calling by MACS2-2.1.1 using broad calling with $p < 0.005$.

### Statistical analysis

GraphPad Prism was used for statistical analysis. All bar graphs display mean ± SD (standard deviation). Significance was assessed by independent two-tailed Student's t test or analysis of variance. $p < 0.05$ was considered statistically significant.

### ImageJ image analysis

ImageJ was used to determine the percentage of the immunostained area. Positive immunofluorescence signals in molar apical regions were first converted to 8-bit binary images and measured using the 'Analyze Particles' function. The derived area was then divided by the total area of apical regions to calculate the percentage of positive immunostaining.

### Data availability

The GEO accession number for the RNA sequencing and ChIP sequencing data reported in this paper is GSE131684.

## Acknowledgements

We thank Julie Mayo, Bridget Samuels and Linda Hattemer for critical reading of the manuscript. We acknowledge USC Libraries Bioinformatics Service for assisting with data analysis. The bioinformatics software and computing resources used in the analysis are funded by the USC Office of Research and the Norris Medical Library. This study was supported by grants from the National Institute of Dental and Craniofacial Research, National Institutes of Health (R01 DE025221 and R37 DE012711).

## Additional information

### Funding

| Funder | Grant reference number | Author |
|---|---|---|
| National Institute of Dental and Craniofacial Research | R01 DE025221 | Yang Chai |
| National Institute of Dental and Craniofacial Research | R37 DE012711 | Yang Chai |

The funders had no role in study design, data collection and interpretation, or the decision to submit the work for publication.

### Author contributions

Junjun Jing, Data curation, Formal analysis; Jifan Feng, Jingyuan Li, Jinzhi He, Thach-Vu Ho, Jiahui Du, Data curation; Xia Han, Software; Xuedong Zhou, Mark Urata, Project administration; Yang Chai, Resources, Supervision

### Author ORCIDs

Junjun Jing (iD) https://orcid.org/0000-0001-5745-5207
Yang Chai (iD) https://orcid.org/0000-0003-2477-7247

### Decision letter and Author response

Decision letter https://doi.org/10.7554/eLife.46426.028
Author response https://doi.org/10.7554/eLife.46426.029

## Additional files

### Supplementary files

• Transparent reporting form
DOI: https://doi.org/10.7554/eLife.46426.020

### Data availability

The GEO accession number for the RNA sequencing and ChIP sequencing data reported in this paper is GSE131684.

The following dataset was generated:

| Author(s) | Year | Dataset title | Dataset URL | Database and Identifier |
|---|---|---|---|---|
| Junjun Jing, Jifan Feng, Jingyuan Li, Jinzhi He, Thach-Vu Ho, Xuedong Zhou, Mark Urata, Yang Chai | 2019 | Antagonistic interaction between Ezh2 and Arid1a coordinates root patterning and development via Cdkn2a in mouse molars | https://www.ncbi.nlm.nih.gov/geo/query/acc.cgi?acc=GSE131684 | NCBI Gene Expression Omnibus, GSE131684 |

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
