## [Decision Letter]

Thank you for submitting your article "Antagonistic interaction between Ezh2 and Arid1a coordinates dental root patterning via Cdkn2a" for consideration by *eLife*. Your article has been reviewed by three peer reviewers, and the evaluation has been overseen by Marianne Bronner as the Senior and Reviewing Editor. The reviewers have opted to remain anonymous.

The reviewers have discussed the reviews with one another and the Reviewing Editor has drafted this decision to help you prepare a revised submission.

Summary:

Rodent tooth has been a traditional model for pattern formation during vertebrate development. In the manuscript, the authors provide the first evidence of how epigenetic mechanisms regulate tooth root formation. Overall, the study is well designed and the hypothesis supported by solid in vivo evidence based on numerous transgenic mouse models. The major findings have general impact to the field of developmental biology. However, several issues need to be addressed as described below.

Essential revisions:

1) Molar dental epithelium diaphragm was shown to be arrested in the *Osr2-Cre;Ezh2^f/f^*mutant mice. However, only mesenchyme proliferation or apoptosis was analyzed (Figure 3). What was the impact upon the epithelium proliferation? What was the cause of epithelium recession? In regards to loss or delay of alveolar bone (Figure 3 alveolar bone), it is not clear if this is primary to the tooth root defects or secondary as this was not determined. Some detailed discussion on this should be added in lieu of the brief short paragraph in the Discussion section. Loss of the PDL is due to mesenchyme ablation of Ezh2, but could they be linked?

2) The authors showed that maxillary molars presented similar root fusion phenotypes in the *Osr2-Cre;Ezh2^f/f^*mutant mice. Were the maxillary molar phenotypes also rescued in the *Osr2-Cre;Ezh2^f/f^; Arid^f/+^* mutant mice? What are the phenotypes of *Osr2-Cre;Ezh2^f/f^; Arid^f/f^* with Arid being completely knocked out? The observed rescue by a single mutant allele of Arid1a is fascinating but the authors should state whether they observe a phenotype in Arid1a -/-roots.

3) In Figure 4, type I collagen was used as a marker for PDL, which is not appropriate. Type I collagen labels both PDL and alveolar bone. It would be better to use periostin as a marker to analyze the PDL defects. It is not clear if the same molar (first or second is compared in G vs. I) or is it the orientation of the section that is confusing? If so please orient the section so they all look the same.

4) In Figure 7, expression of Cdkn2a in *Osr2-Cre;Ezh2^f/f^; Arid^f/+^* mutant mice was only analyzed with immunohistochemical staining, which is not sufficient. It will be necessary to perform either western blot or real-time PCR to confirm.

5) Since the study focusses on epigenetic modifications mediated by Ezh2 it would be interesting to investigate the whole genome status of H3K27Me3 binding sites in wildtype, Ezh2 and Arid1a rescued root mesenchyme. Although I appreciate this may take some time I believe the authors should be given the opportunity to add this data since it will greatly increase the significance of the study. For example, it will be very interesting to know the epigenetic status of the Eda and Arid1a loci.

---

## [Author Response]

Essential revisions:1) Molar dental epithelium diaphragm was shown to be arrested in the Osr2-Cre;Ezh2^f/f^ mutant mice. However, only mesenchyme proliferation or apoptosis was analyzed (Figure 3). What was the impact upon the epithelium proliferation? What was the cause of epithelium recession? In regards to loss or delay of alveolar bone (Figure 3 alveolar bone), it is not clear if this is primary to the tooth root defects or secondary as this was not determined. Some detailed discussion on this should be added in lieu of the brief short paragraph in the Discussion section. Loss of the PDL is due to mesenchyme ablation of Ezh2, but could they be linked?

We have checked the epithelial proliferation in *Osr2-Cre;Ezh2^fl/fl^*mutant molars and found that the proliferation activity in the epithelium was also compromised in *Osr2-Cre;Ezh2^fl/fl^* mutant molars (see Author response image 1). Because *Osr2-Cre* is only active in the dental mesenchyme, the epithelial phenotype is secondary to the defect in the mesenchyme. We have added further information in the Discussion section about the potential relationship between periodontal tissue, including alveolar bone and PDL, and the furcation defect:

*“Osr2-Cre* is active in the dental mesenchyme including both the dental follicle and dental papilla. The dental follicle gives rise to periodontal tissues such as alveolar bone and PDL, which have a critical impact on tooth root development and tooth eruption (Takahashi et al., 2019), implying the interaction between periodontal tissue and tooth root development. In our study, loss of alveolar bone and PDL in the mouse molar correlated with the observed root furcation defect. However, whether the alveolar bone and PDL defects are primary malformations or the result of root furcation defect still needs to be further investigated.”

2) The authors showed that maxillary molars presented similar root fusion phenotypes in the Osr2-Cre;Ezh2^f/f^ mutant mice. Were the maxillary molar phenotypes also rescued in the Osr2-Cre;Ezh2^f/f^; Arid^f/+^ mutant mice? What are the phenotypes of Osr2-Cre;Ezh2^f/f^; Arid^f/f^ with Arid being completely knocked out? The observed rescue by a single mutant allele of Arid1a is fascinating but the authors should state whether they observe a phenotype in Arid1a-/- roots.

Our microCT data has shown that the furcation phenotype of maxillary molars in *Osr2-Cre;Ezh2^fl/fl^*mice is also rescued in *Osr2-Cre;Ezh2^fl/fl^;Arid^f/+^* mutants (see Author response image 2). The *Osr2-Cre;Arid1a ^fl/fl^*and *Osr2-Cre;Ezh2^fl/fl^;Arid1a^fl/fl^*mutant mice die shortly after birth, so we cannot observe the tooth root phenotype. Ablation of *Arid1a* in the mouse embryo will lead to developmental arrest at E6.5 (Gao et al., 2008), therefore, it is also not feasible for us to study the tooth root development in Arid1a-/- mice.

**Author response image 2. respfig2:** 

3) In Figure 4, type I collagen was used as a marker for PDL, which is not appropriate. Type I collagen labels both PDL and alveolar bone. It would be better to use periostin as a marker to analyze the PDL defects. It is not clear if the same molar (first or second is compared in G vs. I) or is it the orientation of the section that is confusing? If so please orient the section so they all look the same.

We appreciate this suggestion. We have replaced the type I collagen with periostin staining and changed the orientation of Figure 4I.

4) In Figure 7, expression of Cdkn2a in Osr2-Cre;Ezh2^f/f^; Arid^f/+^ mutant mice was only analyzed with immunohistochemical staining, which is not sufficient. It will be necessary to perform either western blot or real-time PCR to confirm.

We have performed the qPCR of Cdkn2a in control, *Osr2-Cre;Ezh2^fl/fl^*and *Osr2-Cre;Ezh2^fl/fl^;Arid^fl/+^* mice. The result is consistent with the RNAscope data (see Author response image 3).

**Author response image 3. respfig3:** 

5) Since the study focusses on epigenetic modifications mediated by Ezh2 it would be interesting to investigate the whole genome status of H3K27Me3 binding sites in wildtype, Ezh2 and Arid1a rescued root mesenchyme. Although I appreciate this may take some time I believe the authors should be given the opportunity to add this data since it will greatly increase the significance of the study. For example, it will be very interesting to know the epigenetic status of the Eda and Arid1a loci.

We collected more than one hundred mouse pups at PN 3 days to perform the CHIP sequencing of H3K27Me3 in root mesenchyme of the control molars. Unfortunately, although we appreciate this suggestion we were unable to do this for the compound mutants because we could not collect enough DNA within the two-month revision time frame. We have shown that the Hox genes and Cdkn2a are in the H3K27Me3 binding sites, in consistent with our RNA sequencing data. However, there are no binding sites in the Eda and Arid1a loci (see Author response image 4). An extensive list of H3K27Me3 binding sites in the root mesenchyme is available.

**Author response image 4. respfig4:**